# Alternative Proteins as a Source of Bioactive Peptides: The Edible Snail and Generation of Hydrolysates Containing Peptides with Bioactive Potential for Use as Functional Foods

**DOI:** 10.3390/foods10020276

**Published:** 2021-01-30

**Authors:** Maria Hayes, Leticia Mora

**Affiliations:** 1Teagasc Food Research Centre, Food BioSciences Department, Ashtown, Dublin 15, Ireland; 2Instituto de Agroquímica y Tecnología de Alimentos, Burjassot CSIC, 46980 Valencia, Spain; lemoso@iata.csic.es

**Keywords:** edible garden snail, *Helix aspersa*, protein, hydrolysis, Alcalase^®^, heart health, angiotensin-converting enzyme, mass spectrometry, sustainable protein

## Abstract

Members of the Phylum Mollusca include shellfish such as oysters and squid but also the edible garden snail known as *Helix aspersa*. This snail species is consumed as a delicacy in countries including France (where they are known as petit-gris), southern Spain (where they are known as Bobe), Nigeria, Greece, Portugal and Italy but is not a traditional food in many other countries. However, it is considered an excellent protein source with a balanced amino acid profile and an environmentally friendly, sustainable protein source. The aim of this work was to develop a different dietary form of snail protein by generating protein hydrolysate ingredients from the edible snail using enzyme technology. A second aim was to assess the bioactive peptide content and potential health benefits of these hydrolysates. *H. aspersa* hydrolysates were made using the enzyme Alcalase^®^ and the nutritional profile of these hydrolysates was determined. In addition, the bioactive peptide content of developed hydrolysates was identified using mass spectrometry. The potential heart health benefits of developed snail hydrolysates were measured in vitro using the Angiotensin-I-converting Enzyme (ACE-1; EC 3.4.15.1) inhibition assay, and the ACE-1 inhibitory drug Captopril© was used as a positive control. The generated *H. aspersa* hydrolysates were found to inhibit ACE-1 by 95.60% (±0.011) when assayed at a concentration of 1 mg/mL (*n* = 9) compared to the positive control Captopril© which inhibited ACE-1 by 96.53% (±0.0156) when assayed at a concentration of 0.005 mg/mL (*n* = 3). A total of 113 unique peptide sequences were identified following MS analysis with peptides identified ranging from 628.35 Da (peptide GGGLVGGI—protein accession number sp|P54334|XKDO_BACSU) to 2343.14 Da (peptide GPAGVPGLPGAKGDHGFPGSSGRRGD—protein accession number sp|Q7SIB2|CO4A1_BOVIN) in size using the BIOPEP-UWM database.

## 1. Introduction

Snails from the genus *Helix* (*H. aspersa*, *H. pomatia*, *H. lucorum* or Turkish snail) are terrestrial gastropods commonly used as niche food sources in some European and African countries. In other countries such as China and Japan they are commonly consumed in the breeding and capture forms [1]. *H. aspersa* is known as *H. aspersa maxima*, which is a hermaphrodite snail grown in several temperate climatic regions.

Snail meat is low in cholesterol and fat and Bazán [2] previously reported that it contains 10 mg cholesterol/10 g of meat. There is an increasing demand for alternative protein sources and a requirement for cheaper and more sustainable proteins to feed the growing global population. Snail protein is considered nutritious with an excellent amino acid profile. Snails can be produced in humid countries and European climates generally suit production of *H. aspersa.* However, the consumption of snails is not preferred in several countries in Western Europe and the USA. Despite this, global consumption is 400,000–450,000 tons per year, of which only 15% comes from snail breeding units, while the remaining 85% comes from snails collected from nature [3].

Snails are considered a delicacy and snail meat is usually consumed as an appetizer and prepared in butter. The snail shell is discarded as waste. Several reports exist concerning the nutritional quality of snails [4,5,6]. In addition, there are some reports in the literature regarding the potential benefits of snail meat and eggs. Górka and colleagues [7] recently reported the nutrient and antioxidant properties of *H. aspersa maxima* eggs. Estimated global annual consumption of snail meat is predicted to increase five-fold in the next twenty years [7]. However, there are no studies on the development of snail protein hydrolysates to date as a method to make snail protein more stable and accessible to consumers. Due to the high nutritional value of snail meat products, spoilage sets in after one or two days after harvesting, and preservation methods are required. Hydrolysis can also improve and increase the health benefits of snail proteins.

Hydrolysates consist of short-chains of amino acids known as bioactive peptides. Bioactive peptides are encrypted within the sequence of a parent protein and may be released by hydrolysis using enzymes or acids or by fermentation with lactic acid bacteria (LAB) [8]. The majority of the biologically active peptides identified to date can be found in the BIOPEP-UWM database [9] and include antimicrobial, dipeptidyl peptidase-IV (DPP-IV; EC 3.4.14.5), angiotensin-I-converting enzyme (ACE-I; EC 3.4.15.1) and prolyl endopeptidase (PEP; EC 3.4.21.26)-inhibitory peptides, as well as antioxidant peptides) [10]. Recently, protein hydrolysates were developed from the spotted Babylon snail *Babylonia areolata* using pepsin and pancreatin and the antioxidant activity of the resultant hydrolysate was assessed using the 2,2-Diphenyl-1-picrylhydrazl (DPPH) radical scavenging activity and 2,2′-Azino-bis-3-ethylbenzothiazoline-6-sulphonic acid (ABTS) radical scavenging activity assays as well as a cellular antioxidant assay [11]. Two antioxidant peptides with the amino acid sequences HTYHEVTKH and WPVLAYHFT were identified using MS and WPVLAYHFT had higher radical scavenging activities likely due to the hydrophobic nature of the amino acids within its structure.

The aim of this work was to develop snail protein hydrolysates using the common garden snail and the proteolytic enzyme Alcalase^®^ and to determine yields, nutritional profiles and assess the potential health benefits of this new form of snail protein.

## 2. Materials and Methods

### 2.1. Chemical and Substrate Materials

*Helix aspersa maxima* by-product snails were supplied by Mr Stephen Ryan, Tuam, Co. Galway (Ireland) who farms these snails. By-product snails are those considered to be under-sized and not suitable for market. By-product *Helix aspersa maxima* were carefully washed using running water prior to placement on ice and subsequent freezing at −80 °C. Alcalase^®^ 2.4 L FG was kindly supplied by Novozymes (Bagsvaerd, Denmark). The ACE-I inhibition assay kit was supplied by NBS Biologicals Ltd. (Cambridgeshire, England, UK). The positive control Captopril© was purchased from Sigma-Aldrich (Sigma-Aldrich, Dublin, Ireland). All other chemicals used were of analytical grade.

### 2.2. Generation of Snail Protein Hydrolysates

Generation of snail protein hydrolysates was performed according to the method of Hamid et al. [12] with some modifications. Briefly, 44 g of cleaned, whole by-product *H. aspersa maxima* snail was ground using a Pestle and Mortar and to this 130 mL of distilled deionised water was added. Native enzymes present in the mixture were heat-deactivated by heating the mixture to 85 °C for 15 min in a water bath followed by cooling to room temperature. Following this, the pH was adjusted to pH 8 using 1 M NaOH. The mixture was placed in a shaking incubator at 45 °C and 1.76 mL of Alcalase^®^ enzyme (Sigma Aldrich, Dublin, Ireland) was added to the mixture in an enzyme to substrate ratio of 4%. The temperature was maintained at 45 °C, with shaking at 150 rpm for 180 min. The pH was kept constant at the optimum for Alcalase^®^ using 0.1 M NaOH. After this time, the hydrolysates were heat-deactivated by heating to 90 °C for 15 min. The mixtures were cooled and subsequently centrifuged at 5000 rpm, 4 °C for 20 min after which time the supernatant was recovered, frozen at −80 °C and subsequently freeze-dried along with the shell fraction. The degree of hydrolysis (DH) was calculated using the pHstat technique after 180 min according to the Adler-Nissen method [13].

### 2.3. Freeze-Drying

The supernatant containing peptides was frozen, freeze-dried using an industrial-scale FD 80 model freeze-drier (Cuddon Engineering, Christchurch, New Zealand) and stored at −20 °C until further use. The yield of hydrolysates was calculated based on the percentage of supernatant recovered on a dry weight basis.

### 2.4. Proximate Analysis

The yield of stabilised hydrolysate products was calculated after freeze-drying and weighing the samples, and is expressed as a percentage of the product in total mass of initial snail hydrolysate.

#### 2.4.1. Protein

The total protein content of the samples was determined using the Dumas combustion method using a LECO FP328 Protein analyser (LECO Corp., Saint Joseph, MI, USA), according to the Association of Official Analytical Chemists (AOAC) method 992.15 [14]. The conversion factor of 6.25 was used to convert total nitrogen to protein.

#### 2.4.2. Ash

The ash content was determined gravimetrically, as previously described by Kolar [15].

#### 2.4.3. Lipid Content

The total fat content was determined gravimetrically using an Ankom XT15 Extractor (Ankom Technology, Macedon, NY, USA) for lipid extraction, after previous acid hydrolysis using Ankom HCI Hydrolysis System according to manufacturers’ operating manual and as previously described [16].

#### 2.4.4. Water Activity

The water activity (a_w_) of all samples was measured using an AquaLab Lite meter (Decagon Devices Inc., Munich, Germany). Approximately 0.25 g of finely powdered sample was placed in the water activity metre and aw and the temperature was recorded.

### 2.5. ACE-I Inhibition Assay for Potential to Reduce Blood Pressure

The ACE-I inhibition assay was carried out according to the manufacturer’s instructions and as described previously [16]. In brief, 20 μL of each sample aqueous solution at a concentration of 1 mg/mL was added to 20 μL substrate and 20 μL enzyme working solution in triplicate. Captopril was used as a positive control. Samples were incubated at 37 °C for 1 h. A 200 μL aliquot of indicator working solution was then added to each well, and subsequent incubation at room temperature was carried out for 10 min. Absorbance at 450 nm was read using a FLUOstarOmega microplate reader (BMG LABTECH GmbH, Offenburg, Germany). The percentage of inhibition was calculated using the following equation:% ACE-I inhibition = 100% Initial activity−Inhibitor × 100/100% Initial activity(1)

### 2.6. LC-MS/MS Analysis

Peptides were analysed in a mass spectrometer nanoESI qQTOF (6600 plus TripleTOF, AB SCIEX, Framingham, MA, U.S.A.) using liquid chromatography and tandem mass spectrometry (LC–MS/MS). A total of 1 μL of snail hydrolysate was loaded onto a trap column (3 µ C18-CL 120 Ᾰ, 350 μm × 0.5 mm; Eksigent) and desalted with 0.1% TFA (trifluoroacetic acid) at 5 µL/min during 5 min. The peptides were then loaded onto an analytical column (3 µ C18-CL 120 Ᾰ, 0.075 × 150 mm; Eksigent) equilibrated in 5% acetonitrile 0.1% FA (formic acid). Elution was carried out with a linear gradient from 7 to 45% B in A for 20 min, where solvent A was 0.1% FA and solvent B was ACN (acetonitrile) with 0.1% FA) at a flow rate of 300 nL/min. The sample was ionised in an electrospray source Optiflow < 1 μL Nano applying 3.0 kV to the spray emitter at 200 °C. Analysis was carried out in a data-dependent mode. Survey MS1 scans were acquired from 350 to 1400 m/z for 250 ms. The quadrupole resolution was set to ‘LOW’ for MS2 experiments, which were acquired from 100 to 1500 m/z for 25 ms in ‘high sensitivity’ mode. The following switch criteria were used: charge: 1+ to 4+; minimum intensity; 100 counts per second (cps). Up to 50 ions were selected for fragmentation after each survey scan. Dynamic exclusion was set to 15 s. The system sensitivity was controlled by analysing 500 ng of K562 protein extract digest (SCIEX); in these conditions, 2260 proteins were identified (FDR < 1%) in a 45 min gradient.

### 2.7. Data Analysis of MS/MS Results

Protein Pilot v 5.0. (SCIEX) default parameters were used to generate peak list directly from 6600 plus TripleTOF wiff files. The Paragon algorithm [17] of ProteinPilot v 5.0 was used to search different databases with the following parameters: Uniprot Mollusca database (201,001, 340,255 proteins) and none enzyme. Peptides were identified with a confidence of ≥90%.

### 2.8. In Silico Identification of Novel Peptides

Peptides identified in Table 1 were compared to 4132 bioactive peptides previously reported in the literature [18] and those found in BIOPEP-UWM http://www.uwm.edu.pl/biochemia/index.php/en/biopep (accessed on the 21 December 2020) [19].

## 3. Results

### 3.1. Yields and Proximate Analysis of Snail Protein Hydrolysates

The yield of snail protein hydrolysate recovered following treatment with the proteolytic enzyme Alcalase^®^ was 3.98 g (±0.275) from 44 g of ground snail (9.05% dry weight). The yield of clean shell recovered following hydrolysis was 7.98 g (±0.68) (18.13% dry weight). The degree of hydrolysis was calculated. The protein content of recovered snail protein hydrolysate was 65.49% (±0.90) when measured using the LECO method with a nitrogen conversion factor of 6.25. The lipid content of the developed hydrolysate was 0.51% (±0.31) and the remainder was ash. The a_w_ value of the freeze-dried snail protein Alcalase^®^ hydrolysate was 0.49 (±0.98). All analyses were carried out in triplicate (n = 3). The degree of hydrolysis was calculated as 10.8% (±0.78) using the pH stat technique [13], where *B* (mL) is the volume of NaOH consumed, *N*B is the normality of the NaOH used, 1/α is the average degree of dissociation of the a-amino groups related with the pK of the amino groups at particular pH and temperatures, *M*P (g) is the amount of protein in the reaction mixture, and *h*tot (meq/g) is the sum of the millimoles of individual amino acids per gram of protein associated with the source of protein used in the experiment. Values for *h*tot and 1/α were obtained from the study conducted by Adler-Nissen [13].

### 3.2. Angiotensin Converting Enzyme I Inhibition Assay

The snail Alcalase^®^ hydrolysates showed appreciable ACE-1 inhibition compared to the positive control Captopril©. The percentage ACE-1 inhibition was 95.60 ± 0.011% as compared to 96.53 ± 0.0156% inhibition of ACE-1 by the positive control (Figure 1) when assayed at a concentration of 1 mg/mL as it is shown in Figure 1. Further, the IC_50_ value was calculated as 0.2944 mg/mL, which is comparable to other reported values for marine mollusca hydrolysates (1.50–2.54 mg/mL) [15].

### 3.3. Identification of Proteins and Peptides in Snail Alcalase^®^ Hydrolysates

A total of 113 peptides were identified in the snail Alcalase^®^ hydrolysates. Several (nineteen) of the identified peptides originated from the Alcalase^®^ enzyme, while others (five) were cleaved from the protein actin (Accession number sp|O16808|ACT_MAYDE). Peptides and their parent proteins are shown in Table 1. Several of the peptides shown in Table 1 contain the previously identified di-peptide sequences YG, YA, VY, FG/GF, DF, SF and VW identified previously from *Helix aspersa* as antihypertensive by Drevet [18]. DF is found in four of the peptides and SF in five of the peptides identified in Table 1. These peptides were previously reported to reduce systolic blood pressure in vivo in spontaneously hypertensive rats by 20 mmHg after a single oral administration at doses of 400 and 800 mg·kg^−1^ [18]. 

## 4. Discussion

The yield of protein recovered following the hydrolysis of *H. aspersa* was 9.05% but the protein content of the hydrolysate was significant—65.49% (±0.90). The protein content of the recovered hydrolysate compares favorably with those of marine and animal origin. Previously, Drevet [20] identified seven di-peptides from a hydrolysate of *H. aspersa*. This work identified 113 peptides from an *H. aspersa* hydrolysate generated using the proteolytic enzyme Alcalase^®^. Several of the peptides identified in the hydrolysate contained the di-peptide sequences identified previously within their amino acid sequences and collectively the hydrolysate inhibited the enzyme ACE-1 by 95.60 ± 0.011%. However, the peptides identified were not chemically synthesised and assessed for their ability to inhibit ACE-1 individually. Thus, each peptide is not specifically an ACE-1 inhibitory peptide. However, some of the identified peptides contained shorter sequences with confirmed ACE-1 inhibitory bioactivity. The fact that some regularities were observed previously between the presence of the specific amino acid(s) and the function of a whole peptide sequence has prompted some scientists to find the foundations for the fragmentomic idea of research [21,22,23]. According to the fragmentomic idea, shorter peptide fragments with known bioactivity encrypted in a larger peptide of interest may affect its function, and the presence of ACE-1 inhibitory dipeptides in the parent sequence (understood as the identified longer peptides in this study) could decide the ACE-1 bioactivity of the whole fragment.

None of the peptides identified in this study are reported in the peptide database BIOPEP-UWM http://www.uwm.edu.pl/biochemia/index.php/en/biopep (accessed on the 22 of December 2020) [19]. This database contains over 4132 bioactive peptides and, as it is a comprehensive source of bioactive peptides, no other database was used in this study to assess the novelty of the identified peptides. The usability of in silico methods and databases such as BIOPEP-UWM in supporting the analysis of bioactive peptides derived from food sources was demonstrated previously by authors including Udenigwe [24], Tahir [25] and Minkiewicz and colleagues [19]. In this study, we used it to determine if the identified peptide sequences were novel, but further work to assess the bitterness and further bioactivities of the peptides could be determined using an in silico approach.

Treatment of the garden snail with the enzyme Alcalase^®^ serves as a cheap and effective method of producing protein hydrolysates with ACE-1 inhibitory activities. The ACE-1 inhibitory activity obtained for this hydrolysate is superior to those reported previously in other studies where members of the Mollusca family—green-lipped mussel *Perna canaliculus*—were hydrolysed with Alcalase^®^ and assessed for ACE-1 inhibitory activities [26]. Jayaprakesh and Perera [26] reported ACE-1 inhibition values of ~5% in this study. The hydrolysis of *H. aspersa* generates bioactive ACE-1 inhibitory peptides with potential for the maintenance of normotensive blood pressure in consumers. It also supplies protein in a form that is more convenient than consuming snails in their traditional form.

## 5. Conclusions

In this study, a different dietary form of snail protein by generating peptides from Alcalase enzyme hydrolysis has been described. The obtained peptides have been characterized by mass spectrometry and some of them are potential bioactive peptides that could exert health benefits. In fact, the potential heart health benefits of developed snail hydrolysates were measured in vitro using the ACE-1 inhibitory assay obtaining a significant ACE1 inhibitory activity. Thus, snail Alcalase hydrolysates results an interesting and novel source of bioactive peptides that could exert a protective effect in cardiovascular diseases as well as a sustainable source of high quality protein. 

## Figures and Tables

**Figure 1 foods-10-00276-f001:**
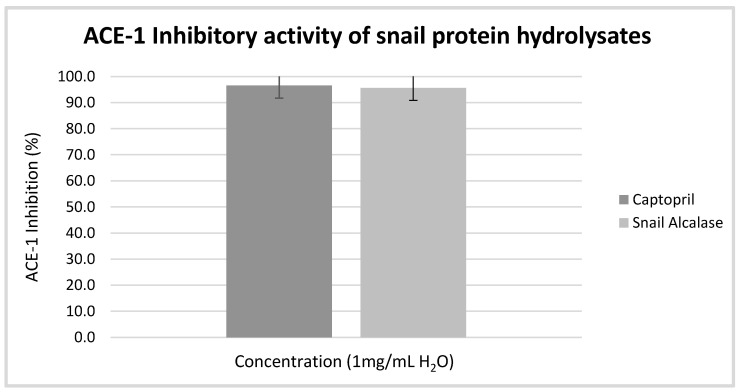
ACE-1 inhibition by snail protein hydrolysate compared to the control Captopril©.

**Table 1 foods-10-00276-t001:** Identification of snail hydrolysate sequences of peptides by mass spectrometry in tandem using the Uniprot Mollusca database (201,001, 340,255 proteins).

Protein Accession Number	Peptide Sequences Identified ^a^	Theoretical Molecular Weight (MW) ^b^
sp|P00780|SUBC_BACLI	AAGNSGSSGNTNTIGYPAKY	1928.89
sp|P00780|SUBC_BACLI	AQTVPYGIPLIK	1298.76
sp|P00780|SUBC_BACLI	AQTVPYGIPLIKADKVQAQ	2039.14
sp|P00780|SUBC_BACLI	ASHPDLNVVGGA	1135.56
sp|P00780|SUBC_BACLI	ASHPDLNVVGGAS	1222.59
sp|P00780|SUBC_BACLI	AVDSNSNRASFS	1253.56
sp|P00780|SUBC_BACLI	AYNTDGNGHGTHVA	1413.59
sp|P00780|SUBC_BACLI	DNTTGVLGVAPSVSLY	1591.81
sp|P00780|SUBC_BACLI	DTGIQASHPDLNVVGGA	1649.80
sp|P00780|SUBC_BACLI	GAVDSNSNRASFS	1310.59
sp|P00780|SUBC_BACLI	GIPLIKADKVQAQ	1379.81
sp|P00780|SUBC_BACLI	ILSKHPNLSAS	1165.65
sp|P00780|SUBC_BACLI	NSSGSGSYSGIVSGIE	1499.67
sp|P00780|SUBC_BACLI	NSSGSGSYSGIVSGIEWATTN	2072.93
sp|P00780|SUBC_BACLI	NTDGNGHGTHVA	1179.49
sp|P00780|SUBC_BACLI	SLGGASGSTAMKQ	1209.57
sp|P00780|SUBC_BACLI	STYPTNTYATL	1230.58
sp|P00780|SUBC_BACLI	STYPTNTYATLN	1344.62
sp|P00780|SUBC_BACLI	VDSNSNRASFS	1182.53
sp|O16808|ACT_MAYDE	AGDDAPRAVFPS	1201.57
sp|O16808|ACT_MAYDE	GFAGDDAPRAVFPS	1405.66
sp|O16808|ACT_MAYDE	GQKDSYVGDEAQSKRGILTL	2164.11
sp|O16808|ACT_MAYDE	KSYELPDGQVITIG	1518.79
sp|O16808|ACT_MAYDE	KSYELPDGQVITIGNE	1761.88
sp|P80057|GSEP_BACLD	SEPIGNTVGYF	1182.56
sp|P80057|GSEP_BACLD	WQHSGPIAISE	1223.59
sp|P02461|CO3A1_HUMAN	GEPGQAGPSGPPGPPGAIGPSGP	1937.91
sp|Q8RB67|DNAJ_CALS4	GEGEPGLRGGPN	1139.52
sp|Q9BXJ0|C1QT5_HUMAN	GEAGPAGPTGPAG	1037.48
sp|B4UGJ6|Y4073_ANASK	EGAVALGAGLALGGSRE	1526.81
sp|O45218|ADAS_CAEEL	LDPANIFASANLIDI	1585.84
sp|Q03A18|ATPB_LACP3	GDPIDGGEAFGP	1130.49
sp|A5WGE2|LPXA_PSYWF	IGNNVILGGNAG	1097.58
sp|O60784|TOM1_HUMAN	SAEGPPGPPSGPA	1119.52
sp|P45586|CU79A_LOCMI	LGGGLGGIGL	812.48
sp|O24385|CPI7_SOLTU	VDDDKDFIPF	1209.56
sp|Q38YA9|GPDA_LACSS	MPITNAIYNVL	1247.66
sp|Q5RF96|SPCS1_PONAB	SGAVAIAFPGLEGPPA	1452.76
sp|P54334|XKDO_BACSU	GGGLVGGI	628.35
sp|Q75JF3|CLCC_DICDI	LIGGLLG	641.41
sp|Q3UHE1|PITM3_MOUSE	AGPSGDSPGSSSR	1142.50
sp|Q8IQG1|MOB2_DROME	LLGGILG	641.41
sp|Q38XN0|MRAY_LACSS	IIGGLIG	641.41
sp|A4R2R1|NST1_MAGO7	NQHYPPGIGPLNAP	1490.72
sp|L0E307|PHQO_PENFE	YLKPVPIVPGLP	1319.79
sp|Q80XI7|VOME_MOUSE	GGLGIGGLL	755.45
sp|Q8BM72|HSP13_MOUSE	VTGVAIQAGIDGGSWP	1526.77
sp|C1A8U3|DAPF_GEMAT	FVKMTGSGNDF	1233.53
sp|P80057|GSEP_BACLD	GYPGDKTAGTQWQHSGPIAISE	2299.09
sp|O42350|CO1A2_LITCT	AGLNGGLGPSGPA	1067.52
sp|P00780|SUBC_BACLI	SHPDLNVVGGA	1064.53
sp|P00780|SUBC_BACLI	AAGNSGSSGNTNTIGYPA	1637.73
sp|Q5FRT2|PROA_GLUOX	LIDAAIAPAL	966.58
sp|P00780|SUBC_BACLI	SKHPNLSAS	939.48
sp|P00780|SUBC_BACLI	APGAGVY	633.31
sp|Q9M3B0|PME34_ARATH	MPVSQIQADIIV	1314.67
sp|Q9Z470|AROA_CORGL	ATAGAIIGLAVDG	1127.62
sp|A1XGT3|TI214_RANMC	LIVLPSLI	866.58
sp|D4GP41|KGSDH_HALVD	GATLVAGGGVPE	1026.53
sp|Q8X226|CAS1_CRYNH	FGLWVLNWI	1147.61
sp|Q9C6V3|AGL86_ARATH	AGAGAGAAPL	754.40
sp|P27483|GRP1_ARATH	GAGGGLGGGHGGGIGGGAGGGSGGGLGGGIGGGAGG	2447.13
sp|P27393|CO4A2_ASCSU	GDDGLPGAPGRPG	1146.54
sp|P00780|SUBC_BACLI	NSSGSGSYSGIVS	1200.53
sp|B2A865|ACCDA_NATTJ	EGGSGGALALTV	1030.53
sp|Q8WXI7|MUC16_HUMAN	PSLLSLPATTSP	1182.65
sp|Q9H7P9|PKHG2_HUMAN	RGGGGGGPR	769.39
sp|Q1HVF7|EBNA1_EBVA8	GAGGGAGAGGGAGAGGGAGAGGGAG	1555.67
sp|Q80XI7|VOME_MOUSE	GGEGGGLGIGGLL	1055.56
sp|Q7W0A6|MUTL_BORPE	AGVPDGAAPDTAYAGEPA	1628.73
sp|Q8IA41|GLT11_DROME	EPILLNNQ	939.50
sp|Q8K4I4|BPIA1_RAT	NGLVGGLLG	798.46
sp|B8ITX3|MCH_METNO	VAEAAGVPL	825.46
sp|Q9LEJ0|ENO1_HEVBR	SIEDPFDQD	1064.43
sp|P86950|SLP2_PINMA	GIGGGGIIGGGPI	1023.57
sp|Q5U9X3|GPC6A_DANRE	IIGGLFPI	828.51
sp|Q9NG98|TOP3A_DROME	GGGGGPGPGPGGG	879.38
sp|O16808|ACT_MAYDE	RVAPEEHPVLL	1258.70
sp|P34804|COL40_CAEEL	SEPGPAGPAGDAGPDGAPGNAGAPGA	2114.91
sp|Q2KIN5|HEM3_BOVIN	WSLNGAETMQ	1136.48
sp|Q9FZC4|FOX1_ARATH	PAGTPKTVLLGRP	1305.78
sp|P12575|FUS_SENDF	IVVMVVIL	884.58
sp|B0TMM7|ADEC_SHEHH	LDALAPLI	824.50
sp|P00780|SUBC_BACLI	SHPDLNVVGGAS	1151.56
sp|P49597|P2C56_ARATH	AGPFRPF	790.41
sp|Q7TZN1|PKNF_MYCBO	TEAPLPIE	868.45
sp|Q6AZY7|SCAR3_HUMAN	GDPGSLGPLGPQ	1094.52
sp|P17140|CO4A2_CAEEL	QDGLPGLPGNKG	1152.58
sp|A5G0G6|LEUC_ACICJ	LGMNPDKLKPGE	1297.67
sp|Q7SIB2|CO4A1_BOVIN	GPAGVPGLPGAKGDHGFPGSSGPRGD	2343.14
sp|Q96MP8|KCTD7_HUMAN	FGDVLNF	810.39
sp|Q8TZZ2|MPTA_PYRFU	EDIALEDMI	1047.48
sp|Q7Z5A4|PRS42_HUMAN	APGPEAGPPL	904.47
sp|O48534|DEXHD_ARATH	VQVGVAINGE	984.52
sp|Q9QY06|MYO9B_MOUSE	DAGLSPGSQGDSK	1217.55
sp|Q4K758|SYM_PSEF5	ITQYFDPE	1011.45
sp|Q9Y4K4|M4K5_HUMAN	SSDPNFMLQ	1038.43
sp|Q8SWH6|Y206_ENCCU	DVPVEEMAVG	1044.48
sp|Q82EX7|DNAJ1_STRAW	GAGGGFGGGI	748.35
sp|B2GDS9|DNLJ_LACF3	AGDIIPE	713.36
sp|Q9Z8N1|PHSG_CHLPN	AIEDIALI	856.49
sp|Q9FCC1|BIOD_STRCO	GAPLLGAVPAGAGS	1136.62
sp|Q6CZR3|GPH_PECAS	IGGDDVIVK	914.51
sp|P0C062|GRSA_BREBE	GGEGLARGYWK	1192.60
sp|P21840|VSM5_TRYBR	GEDQETFHSRFWDQ	1781.73
sp|Q97CT6|METK_THEVO	DTSFGVGFAP	996.46
sp|Q2KJ58|R3GEF_BOVIN	SVGPCKSHRESLGGLPE	1751.86
sp|Q10Y85|RIMO_TRIEI	GTPAYNLPN	945.46
sp|Q4K5F9|PROA_PSEF5	NEVDSSSVMVNASTRF	1741.79
sp|D5AV94|TKT_RHOCB	FVGMEGFGASAPA	1239.56
sp|Q9KPI4|Y2383_VIBCH	AAAQLALGGML	1014.55
sp|Q8SY41|BCAS3_DROME	GLGVQVWAIPANGEAVE	1708.88
sp|Q9DD48|MKRN2_SERQU	GGGGAGGGGAGIGGAGGGP	1239.56
sp|O48928|C77A3_SOYBN	TALAFFISGLIF	1298.73

^a^ The sequence of the peptide identified by the search; ^b^ The theoretical precursor molecular weight for peptide sequence, including modifications.

## Data Availability

Data is contained within the article. The data presented in this study is available in the present article.

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
