# Peer review of "Alternative Proteins as a Source of Bioactive Peptides: The Edible Snail and Generation of Hydrolysates Containing Peptides with Bioactive Potential for Use as Functional Foods"

_foods, 2021, doi:10.3390/foods10020276_

Round 1

Reviewer 1 Report

In my opinion, this paper needs some modifications.

Here are some details:

Title should be changed into: Alternative proteins as a source of bioactive peptides: the edible snail and generation of hydrolysates containing peptides with bioactive potential for use as functional foods (or similar).

Please unify the name of the enzyme (Alcalase). Sometimes it is written with capital letter, sometimes not.

You cite the publication using the name of its author „in the middle” of the text and put the number of the cited reference at the end of the sentence.  Please put the number of the cited reference behind such name.

Example: Górka and collegues [7] but not Górka and collegues……..[7]. Pease double-check the whole manuscript and correct it.

Since 2015, BIOPEP database has been known as BIOPEP-UWM database. Please correct it through the whole paper.

Line 228 - it should be BIOPEP-UWM instead of BIOPEP-UMW.

If you are providing the a.ccession number of protein for the first time, please also provide the name of the database you worked with (line 29)

Figure 1. Lack of the title of it.  I think the second bar should be explained as „snail alcalase hydrolysates”

I think that discussion must be extended. Peptides that were identified in snail hydrolysates were not measured (individually) towards their bioactivity. Thus, in my opinion it cannot be said, that they are novel or they were ACE inhibitors. It was observed that these peptides contained shorter sequences with confirmed ACE inhibitory bioactivity. Such approach was called by some authors as „fragmentomic idea” according to which the presence of bioactive/functional fragments in a „parent sequence” (understood as protein or longer peptide) may decide about the bioactivity of a whole fragment. I think, the discussion would be more valuable if the authors added „few words” about it.

Peptides that were identified in snail hydrolysates were called as „novel” based on BIOPEP-UWM search. Were there any trials for searching for these peptides in other databases? If not, I think that discussion would be more valuable if it contained a fragment about „usability of in silico methods in supporting the analysis/prediction of bioactive peptides derived from foods”.

Lack of Conclusions

Finally:  I find this article interesting for those who search novel sources of proteins being the precursors of bioactive peptides. After the completing this manuscript with some issues concerning the fragmentomic approach as well as the role of in silico methodology in prediction of bioactive peptides in foods, it can be publish in Foods.

Author Response

Dear reviewer, many thanks for your detailed comments. We have provided the response to reviewer comments as a PDF file below. Kind regards, Maria

Reviewer 2 Report

The paper written by the authors Maria Hayes and Leticia Mora, frames new sources of protein. In particular, the study is conducted on snails and bioactive peptides have been identified for the ACE-1 antihypertensive assay. The manuscript is very well written, the experiments are well explained, and a positive control for comparison has also been included. I have no recommendations or requests for the authors. Study has been conducted in an accurate and well-done manner.

The authors could include a paragraph, the investigated bioactivity may be due to the whole list of identified peptides or to only one of those, without the synthesis of these and specific tests it cannot be certain. There are works in the literature that arrive at the identification of unique peptides responsible for bioactivity, to give examples.

Author Response

We would like to thank reviewer 2 for their positive review of our paper. We have provided our response to reviewer comments in the attached PDF document.
